# Preclinical Evaluation of the Multiple Tyrosine Kinases Inhibitor Anlotinib in Leukemia Stem Cells

**DOI:** 10.3390/ph15111313

**Published:** 2022-10-25

**Authors:** Yuelong Jiang, Long Liu, Yirong Jiang, Zhifeng Li, Liying Feng, Xinguo Zhuang, Zhijuan Lin, Qiuling Chen, Guoshu Chen, Jixiang He, Guowei Li, Jie Zha, Bing Xu

**Affiliations:** 1Department of Hematology, The First Affiliated Hospital of Xiamen University and Institute of Hematology, School of Medicine, Xiamen University, Xiamen 361003, China; 2Xiamen Key Laboratory of Diagnosis and Therapy for Hematological Malignancies, Xiamen 361003, China; 3Department of Hematology, Affiliated Dongguan People’s Hospital, Southern Medical University (Dongguan People’s Hospital), Dongguan 523059, China; 4Department of Hematology, Huizhou Municipal Central Hospital, Huizhou 516001, China

**Keywords:** acute myeloid leukemia, leukemia stem cell, anlotinib, JAK, STAT3, chemoresistance

## Abstract

Leukemia stem cells (LSCs) constitute the critical barrier to the cure of acute myeloid leukemia (AML) due to their chemoresistance and immune evasion property. Herein, the role of anlotinib, a multiple tyrosine kinase inhibitor, in killing LSCs and regulating chemoresistance and immune evasion was explored. Anlotinib treatment induced apoptosis of LSC-like cells as well as primary AML LSCs, while sparing the normal mononuclear cells in vitro. Moreover, anlotinib could impair the regeneration capacity of LSCs in the patient-derived leukemia xenograft mouse model. Mechanistically, anlotinib inhibited phosphorylation of c-kit, JAK2/STAT3, and STAT5, and downregulated STAT3 and STAT5 expression. In addition, anlotinib downregulated the anti-apoptotic proteins Bcl-2 and Bcl-xL, and upregulated Bax, thereby enhancing the sensitivity of LSCs to idarubicin in vitro. Intriguingly, anlotinib could also partially rescue the interferon-g production of T cells cocultured with LSCs by downregulating PD-L1 expression. In conclusion, anlotinib showed anti-LSC activity and the potential to enhance the sensitivity to idarubicin and inhibit the immunosuppressive feature of LSCs via JAK2/STAT signaling pathway downregulation in the preclinical study. Our results provided a rational basis for combinatory strategies involving anlotinib and chemotherapy or immunotherapy.

## 1. Introduction

The treatment of acute myeloid leukemia (AML) has been rapidly evolved from conventional chemotherapy to target therapeutics such as FLT3-ITD and IDH2 inhibitors, thereby improving the complete remission rates over the past decade [1,2]. However, novel therapeutics have not significantly reduced the relapse of AML, leading to the failure in converting complete remission to the long-term survival of patients [3]. Thus, preventing relapse remains the main focus of AML management.

Leukemia stem cells (LSCs) are a small population at the ape of the AML differentiation hierarchy, characterized by self-renewal, chemoresistance, and immune evasion [4,5,6]. They are also considered as the source of relapse after chemotherapy [7,8]. Therefore, targeting LSC is a rationally promising strategy to prevent AML relapse [5,6,9,10]. As reported previously, growth factors, such as stem cell factor (SCF), interleukin 3 (IL-3), and granulocyte-colony stimulating factor (G-CSF) play an important role in the maintenance of LSC stemness and survival [4]. Accordingly, phenotypically high expression of growth factor receptors, such as IL-3 receptor (CD123), c-kit, and FMS-related tyrosine kinase 3 (FLT3), or genetic mutations in these receptors or their downstream pathways, such as neuroblastoma RAS viral (v-ras) oncogene homolog (N-RAS), were observed in the LSCs of AML, which incurred the overactivation of their downstream tyrosine kinases [11,12,13]. Such overactivation could be engaged in the proliferation, apoptosis, and immune evasion through multiple signaling pathways, especially JAK/STAT [11,14]. STAT3/5 is the downstream of JAK2 signaling, which regulates the chemo-sensitivity through influencing the balance between pro- and anti-apoptotic proteins, such as Bcl-2 and Bcl-xL, and affecting the expression of immune checkpoint ligands, such as PD-L1 [15,16,17]. Notably, STAT3/5 activation has been dependent on the growth factor signaling in LSCs [11], implying that targeting growth factor receptors and their downstream signaling might be effective for eliminating LSCs. 

Anlotinib is a multiple kinase inhibitor that targets c-kit, platelet-derived growth factor receptor (PDGF-R), fibroblast growth factor receptor (FGF-R), and vascular endothelial growth factor receptor (VEGF-R), and has demonstrated efficacy in non-small cell lung cancer and other cancers with acceptable safety profile [18,19,20]. Additionally, Liang et al. reported that anlotinib could exert its role through inhibition of JAK2 signaling [21]. Moreover, the activation of FGFR1b signaling pathway promotes the resistance to chemotherapy in AML cells [22], and upregulates the expression of PD-L1 on colorectal cancer via JAK/STAT signaling [23]. However, the role of anlotinib in regulating chemoresistance, immune escape properties and the LSC survival is yet to be elucidated. Herein, we demonstrated that anlotinib eliminates LSCs, sensitizes LSCs to idarubicin (IDA), and potentially inhibits the immune escape by blockade of JAK/STAT3/5 signaling.

## 2. Results

### 2.1. Anlotinib Exerts Potential Anti-LSCs Effects by Inhibiting Proliferation and Inducing Apoptosis of LSC-Like Cells In Vitro

The CD34^+^CD38^−^ population in two LSC-like cells (KG-1α and Kasumi-1) was sorted by magnetic-activated cell sorter (MACS). To evaluate the effect of anlotinib on cell proliferation in LSC-like cell lines, we treated CD34^+^CD38^−^ KG-1α and CD34^+^CD38^−^ Kasumi-1 cells with prespecified concentrations of anlotinib for 48 and 72 h, and measured cell proliferation by CCK-8 assay. The results showed that anlotinib treatment inhibited Kasumi-1 and KG-1α proliferation in a dose- and time-dependent manner (Figure 1A,B). To investigate whether cell proliferation blockade was caused by apoptosis or cell cycle arrest, apoptosis, and cell cycle of LSCs after anlotinib treatment were further evaluated. As a result, anlotinib also induced apoptosis of LSCs in a dose- and time-dependent manner (Figure 1C,D). In addition, the cell cycle arrest in G2/M phase was also observed after anlotinib treatment (Figure 1E,F). Therefore, these data indicated that the anti-LSC effects of anlotinib might depend on its cytotoxicity and the capacity of inducing cell cycle arrest in G2/M phase.

### 2.2. Anlotinib Targets Primary LSCs While Sparing Normal Mononuclear Cells Ex Vivo

Although anlotinib could effectively kill LSC-like cells, its exact effect on primary LSCs is yet unclear. Therefore, we treated the isolated mononuclear cells from the bone marrow of 9 AML patients with escalating doses of anlotinib. The clinical characteristics of AML patients were summarized in Table 1. Consistent with the effects in LSC-like cell lines, anlotinib also significantly induced the apoptosis of CD34^+^ AML cells in a dose-dependent manner (Figure 2A). In addition, we further evaluated the adverse effects of anlotinib on normal blood cells. Mononuclear cells from the peripheral blood of healthy donors were treated with anlotinib for 48 h. Intriguingly, even the maximal dose (10 μM) did not show significant cytotoxicity (Figure 2B). CD34^+^CD38^−^LSCs from the bone marrow of AML patients were sorted to evaluate anlotinib on primary LSCs. Similarly, anlotinib treatment also induced the apoptosis of CD34^+^CD38^−^LSCs in a dose–dependent manner. (Figure 2C).

### 2.3. Anlotinib Impairs the Regeneration Capacity of LSCs in Patient-Derived Xenograft Mice Models

The regeneration of leukemia is the feature of LSCs. Whether anlotinib impairs the regeneration capacity of LSCs is yet to be clarified. We generated a patient-derived xenograft mice model to demonstrate the impact of anlotinib on the regeneration capacity of CD34^+^CD38^−^LSCs. The spleen weight in anlotinib group was also significantly less than the control group (Figure 3A,B). Anlotinib pretreatment significantly decreased the engraftment of hCD45^+^ cells in spleen and hCD45^+^CD34^+^CD38^−^ cells in bone marrow of mice at the 4th week (Figure 3C,D). Although LSCs were enriched in the CD34^+^CD38^−^ population, the CD34^+^CD38^+^ population also contained LSCs. Therefore, mononuclear cells of bone marrow from patients with CD34^+^CD38^+^AML cells were used to establish the PDX model. Similarly, pretreatment with anlotinib led to a lower leukemic burden in spleen manifested as lower spleen weight and hCD45^+^ cells in spleen (Figure 3E–G). Moreover, anlotinib pretreatment also hindered the engraftment of hCD45^+^CD34^+^ cells in the bone marrow (Figure 3H).Collectively, anlotinib effectively suppressed the regeneration capacity of LSCs derived from AML patients regardless the phenotype of LSCs.

### 2.4. Anti-LSCs Activity of Anlotinib Might Be Associated with the Inhibition of the JAK-STAT Pathway

The overactivation of JAK/STAT signaling is constitutively increased in AML stem and progenitor cells due to the mutations of RTKs or growth factor stimulation, which is responsible for the self-renewal of LSCs [24]. Whether anlotinib inhibits JAK-STAT pathway is yet to be elucidated. We determined the impact of anlotinib on JAK-STAT pathway on LSC-like cell lines as well as primary CD34^+^CD38^−^LSCs. Anlotinib significantly inhibited the activation of c-kit and JAK2 in both LSC-like cell lines (Figure 4A,B) and primary LSCs (Figure 4C). Moreover, the phosphorylation of downstream transcription factors of JAK2, including STAT3 and STAT5, was inhibited in both LSC-like cell lines (Figure 4A,B) and primary LSCs (Figure 4C) after anlotinib treatment. Notably, the expression of STAT3 and STAT5 was decreased by anlotinib (Figure 4A–C). These results indicated anlotinib inhibits the activation of JAK2/STAT signaling and downregulates STAT3 and STAT5.

### 2.5. Anlotinib Enhances the Sensitivity of LSC-Like Cells to Idarubicin by Regulating Apoptosis-Related Proteins

Chemoresistance is considered as the key feature of LSCs [12]. The overexpression of anti-apoptotic proteins, such as Bcl-2 and Bcl-xL, was associated with chemoresistance. Pro- and anti-apoptotic proteins were reported as the targets of JAK2/STAT signaling in various tumors, such as leukemia and lymphoma [15,25]. Hence, we evaluated the role of anlotinib in regulating these proteins in LSCs. As shown in Figure 5A,B, the upregulation of Bax and downregulation of Bcl-2 and Bcl-xL in Kasumi-1 and KG-1α were observed in the anlotinib group. We further evaluated the role of anlotinib in modulating the sensitivity to idarubicin (IDA). As a result, anlotinib significantly enhances the sensitivity of LSC-like cells to IDA (Figure 5C,D).

### 2.6. Anlotinib Disrupts the Immunosuppressive Effects of LSC-Like Cells on T Cells via Downregulation of PD-L1

Accumulating evidence showed that FGFR, c-kit, and VEGFR signaling are linked to PD-L1 expression through activation of JAK/STAT signaling in several solid tumors, such as colorectal/lung and renal cancer. Whether the inhibition of JAK2/STAT3/5 signaling by anlotinib downregulates PD-L1 expression is yet unclear. Interestingly, a significant decrease in PD-L1 expression (Figure 6A,B) was observed in LSC-like cells after anlotinib treatment. Since the PD-1/PD-L1 axis significantly impaired T cell function, we further demonstrated whether downregulation of PD-L1 disrupted the inhibitory effects of LSCs on T cell function. The coculture of Kasumi-1 with PBMCs from healthy donors demonstrated that the anlotinib treatment partially rescues the interferon-g (IFN-γ) production of CD4^+^T cells (Figure 6D), CD8^+^T cells (Figure 6E), and total CD3^+^T cell compartment (Figure 6C). Moreover, anlotinib did not compromise IFN-g production of T cells without coculture with AML cells (Figure 6C–E).

## 3. Materials and Methods

### 3.1. Cell Line Culture

CD34^+^CD38^−^ population of Kasumi-1 and KG-1α cell lines were stored in our laboratory. Kasumi-1 was cultured in RPMI 1640 (HyClone^TM^, Logan, UT, USA) supplemented with 100 U/mL penicillin and 100 μg/mL streptomycin (1×P/S) and 10% fetal bovine serum (FBS) (HyClone). KG-1α was cultured in IMDM (HyClone) supplemented with 1×P/S and 10% FBS.

### 3.2. Patients and Donor Samples

Primary samples from the bone marrow of AML patients (*n* = 12) and peripheral blood of healthy donors (*n* = 14, 8 for apoptosis assay and 6 for experiments related to T cell function) were collected at the Department of Hematology, First Affiliated Hospital of Xiamen University after patients provided informed consent. This study was carried out in accordance with the Declaration of Helsinki and approved by the Ethics Review Board of First Affiliated Hospital of Xiamen University. The clinical characteristics of patients with AML are summarized in Table 1. The mononuclear cells were isolated by density gradient centrifugation using Ficoll (BD, Franklin Lakes, NJ, USA) and supplemented with RPMI 1640 and 10% FBS for short-term culture. 

### 3.3. Cell Viability Assay

Cell proliferation was determined using a CCK-8 kit (MCE, Shanghai, China). Briefly, (3 × 10^4^ cells/well were plated in 96-well plates containing 100 μL of growth medium and then treated with designated doses of anlotinib (Chia Tai Tianqing Pharmaceutical Group Co., Ltd., Lianyungang, China) for 48 and 72 h, respectively. The CCK-8 agent was added to the well and incubated for 2–4 h in an incubator after treatment. The absorbance was measured at 450 nm on a VERSA max microplate reader (Molecular Devices, Sunnyvale, CA, USA).

### 3.4. Detection of Apoptosis

The apoptosis of LSCs or LSC-like cells was detected by flow cytometry after staining with Annexin V and PI following the manufacturer’s instructions. Briefly, cells exposed to different treatments were harvested and washed with ice-cold phosphate-buffered saline (PBS; Gibco BRL, Rockville, MD, USA) and 1× Annexin V binding buffer and then subjected to Annexin V-FITC/PI staining. Flow cytometry analysis was performed within 1 h after staining. Specific apoptosis was adopted to adjust the variation in basal levels of spontaneous cell death according to the following formula: (% apoptosis in treated cells − % apoptosis in untreated cells)/(1 − % apoptosis in untreated cells) to analyze the potential correlation between patients’ characteristics and ex vivo efficacy of anlotinib.

### 3.5. Ex-Vivo Evaluation of Cytokine Production of T Cells

Peripheral blood mononuclear cells (PBMCs) from healthy donors were isolated by density gradient centrifugation. The PBMCs were cultured in RPMI 1640 medium (GIBCO) containing 10% fetal bovine serum and interluekin-2 (100 U/ml) with Kasumi-1 cells at the ratio 1:1. After anlotinib(1 μM) or PBS treatment for 3 days, PBMCs were stimulated with phorbol 12-myristate 13-acetate (PMA)/ionomycin (100 ng/mL and 1 μg/mL) as well as Golgiplug (BD Pharmingen, San Diego, CA, USA) for 4 h. Cells were harvested for surface staining by CD3-APC-Cy7 (UCHT1, Biolegend Corp., San Diego, CA, USA) CD4-APC (SK3, Biolegend), CD8-PE-CY7(RPA-T8,Invitrogen Corp., Waltham, MA, USA) and intracellular staining with IFN-γ-PE (Biolegend) and then subjected to analysis by flow cytometry.

### 3.6. Western Blot Analysis

An equivalent of 2 × 10^5^ cells/well were treated with anlotinib at indicated concentrations for 12 h. The apoptosis of cells was tested to control the apoptotic rates to <10% before immunoblot assays. Subsequently, the cells were lysed in RIPA buffer containing protease inhibitor cocktails. The protein level of each sample was determined using a BCA protein Assay (Pierce, Thermo Fisher Scientific, Eugene, OR, USA). An equivalent of 20 μg protein/lane was separated by SDS-PAGE and transferred to a PVDF membrane (Millipore Corp., Burlington, MA, USA). Then, the membrane was blocked by 5% non-fat milk and probed with primary antibodies: anti-c-kit (18691-1-AP, Proteintech Corp., Suite, USA) PD-L1 (M1H1,Invitrogen Corp., Waltham, MA, USA)and anti-JAK-2 (3230), anti-Jak-2 (3230), anti-Phospho-JAk-2 (8082), anti-Stat3 (30835), anti-Stat3 (9339S), anti-Phospho-Stat3 (4113S), anti-Stat5 (94205S), anti-Phospho-Stat5 (4322S), anti-Bcl-2 (4223S), anti-Bcl-xL (2764S), anti-Bax (5023S), anti-Phospho-c-kit (3073S), and anti-GAPDH (D16H11) from Cell Signaling Technology (Danvers, MA, USA). After incubation with horseradish peroxidase (HRP)-conjugated secondary antibodies, the immunoreactive signals were detected using an enhanced chemiluminescence (ECL) substrate (GE Healthcare, Chicago, IL, USA) and visualized using the Amersham Imager 600 (AI600, GE Healthcare). 

### 3.7. AML Patient-Derived Xenograft (PDX) Mice Models

Two patient-derived xenograft mice models were applied in this study to evaluate the impact of anlotinib on the regeneration capacity of CD34^+^CD38^−^LSCs and CD34^+^ LSCs respectively. The first patient-derived xenograft model was generated using CD34^+^CD38^−^LSCs sorted from bone marrow of an AML patient while the other one was built using bone marrow mononuclear cells from another patient with CD34^+^CD38^+^AML cells. CD34^+^CD38^−^LSCs or bone marrow mononuclear cells were pretreated with anlotinib (2.5 μM) or vehicle (PBS), respectively, for 12 h in vitro. To exclude the impact of apoptotic cells on the engraftment of LSCs, we detected percentages of apoptotic cells in each group. Then 1 × 10^6^ live cells were injected into the tail vein of NOD-Prkdc^−/−^IL2rg^−/−^ (NSG) mice (DMO Ltd., Beijing, China) after 1 Gy irradiation. In order to confirm the engraftment, hCD45^+^ cells were detected in the peripheral blood. At the 4th week, the NSG mice were sacrificed to analyze the tumor burden with human CD45-FITC (clone HI30, Biolegend) and murine CD45-BV421 (clone 30-F11, Biolegend) in spleen and evaluate LSCs with CD34-APC antibodies (clone 581, Biolegend) and/or CD38 (clone HB7, Biolegend) in bone marrow using flow cytometry.

### 3.8. Statistical Analysis

The data are expressed as the mean ± standard error of mean (SEM) for at least three independent experiments. Student’s *t*-test or/and Mann–Whitney tests were applied in the two-group comparison according to the distribution of the data. Wilcoxon test were used to compare T cell function in different groups. Mann–Whitney test was performed in vivo studies. *p* < 0.05 indicates statistical significance. All statistical analyses were performed using SPSS 22.0 software (La Jolla, CA, USA). 

## 4. Discussion

The current research demonstrated that anlotinib has an anti-LSC activity associated with the inhibition of JAK2-STAT3/5 signaling. Additionally, anlotinib enhances the sensitivity to IDA and compromises the immune-evasion potential of LSCs by downregulating the anti-apoptotic proteins and immune checkpoint receptors, respectively. The current results provide a strong scientific rationale for the application of anlotinib in AML.

Overactivation of JAK2-STAT signaling is caused by various cytokines, growth factors, or mutation of receptor tyrosine kinases, such as c-kit, FLT3, or PDGFR, associated with self-renewal of LSCs [26]. Anlotinib induces the apoptosis of LSCs and also inhibits the regeneration of LSCs in vivo. Compared to the JAK2 inhibitor, anlotinib exhibits mechanistic advantages by the inhibition of JAK2-STAT3/STAT5 activation and STAT3 and STAT5 expression. Such advantages might be associated with the inhibition of multiple targets, such as c-kit and PDGFR, by anlotinib, which activates other oncogenic signaling pathways, including Ras/MAPK, PI3K/AKT, NF-kB, and JAK2 [14]. However, further studies are warranted to investigate the underlying mechanisms. Additionally, hematopoiesis toxicity is reported as one of the main adverse effects of JAK2 inhibitor in clinical practice [27], which limits its wide application. However, clinical data in non-small cell lung cancer [18,19] and our laboratory data showed that anlotinib does not exert severe hematological and hepatic adverse effects.

IDA combined with cytarabine is the frontline therapy for AML patients [24]. Bcl-2 interacts with pro-apoptotic BH3-only proteins, such as Bax, BIM, and BID, and subsequently blocks the intrinsic apoptotic pathway [28]. Bcl-2 upregulation in leukemia cells contributes to the resistance of IDA or cytarabine in AML [29]. Reportedly, Bcl-2 expression could be regulated by JAK2-STAT3/5 pathway. Although several studies have demonstrated that the inhibition of JAK2 signaling alone or multiple tyrosine kinases upstream of STAT3/5 in reducing the maintenance and self-renewal of LSCs, their impact on chemosensitivity is not yet reported. In other settings, such as lymphoma and pulmonary fibrosis, downregulation of Bcl-2 expression by inhibition of JAK signaling was demonstrated [25,30]. Our data also showed that anlotinib significantly decreases Bcl-2 expression and upregulates Bax expression in LSC-like cell lines and enhances the sensitivity of LSCs to IDA.

PD-L1/PD-1 axis plays a critical role in the immune evasion of AML by producing a potent immunosuppressive effect on T cells [31]. An increase in the number of dysfunctional T cells with the upregulation of immune checkpoint receptors, such as PD-1, TIGIT, and Tim-3 was observed in AML patients [32,33]. Reportedly, FGFR upregulates PD-L1 expression by activating the JAK/STAT signaling pathway in colorectal carcinoma [23]. Liu et al. also demonstrated that anlotinib downregulates PD-L1 expression on vascular endothelial cells in lung cancer [34]. In addition, whether anlotinib modulates the expression of PD-L1 on tumor cells remains unclear. Our results showed that anlotinib significantly downregulates PD-L1 expression on LSCs. Notably, anlotinib treatment partially rescues the dysfunction of T cells cocultured with LSCs without an impact on normal T cells.

There are some limitations in the current study. Compared with higher activity in JAK2 signaling in LSCs, the lower activity of JAK2 signaling in CD34^+^ stem cells might be more easily compensated by other pathways [11]. This observation might contribute to the selective cytotoxicity of anlotinib against LSCs but not normal CD34^+^ stem cells, which needed to be verified in the future. Additionally, further studies are warranted to investigate the mechanism about the downregulation of anti-apoptotic proteins by anlotinib in LSCs.

In conclusion, our preclinical findings provide evidence for the clinical implementation of anlotinib as a novel approach to eliminate LSCs. Given the role of anlotinib in regulating the sensitivity to IDA and immune evasion and its low adverse effects, combining chemotherapy such as IDA and cytarabine, or immunotherapy such as immune checkpoint inhibitors might be a promising strategy to improve the outcome of AML.

## Figures and Tables

**Figure 1 pharmaceuticals-15-01313-f001:**
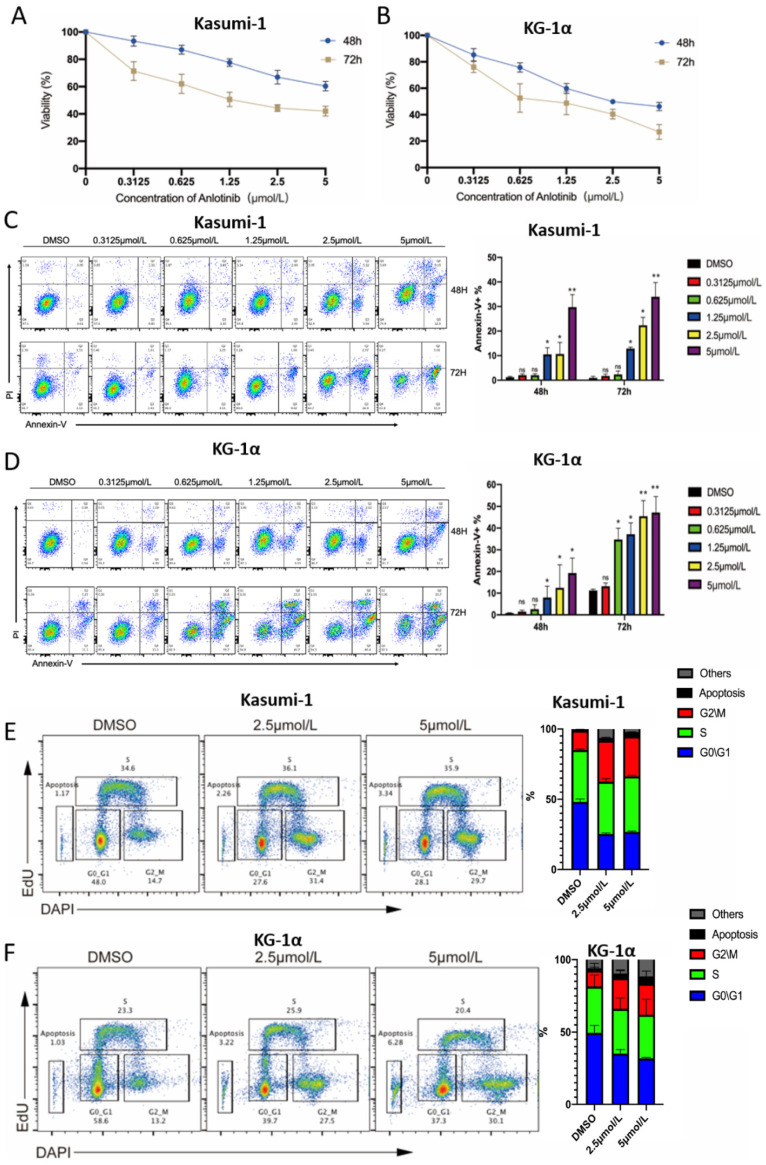
Anti-LSC activity of anlotinib in vitro. (**A**,**B**) Cell proliferation was detected by CCK-8 assay of CD34^+^CD38^−^ Kasumi-1 cells (**A**) and KG-1α cells (**B**) after treatment with the indicated concentrations of anlotinib for 48 and 72 h. (**C**,**D**) Apoptosis of CD34^+^CD38^−^ Kasumi-1 cells (**C**) and KG-1α cells (**D**) after anlotinib treatment with the indicated concentrations of anlotinib for 48 and 72 h. (**E**,**F**) Cell cycle detection of CD34^+^CD38^−^ Kasumi-1 cells (**E**) and KG-1α cells (**F**) after anlotinib treatment with the indicated concentrations of anlotinib for 6 h. * *p* < 0.05, ** *p* < 0.01.

**Figure 2 pharmaceuticals-15-01313-f002:**
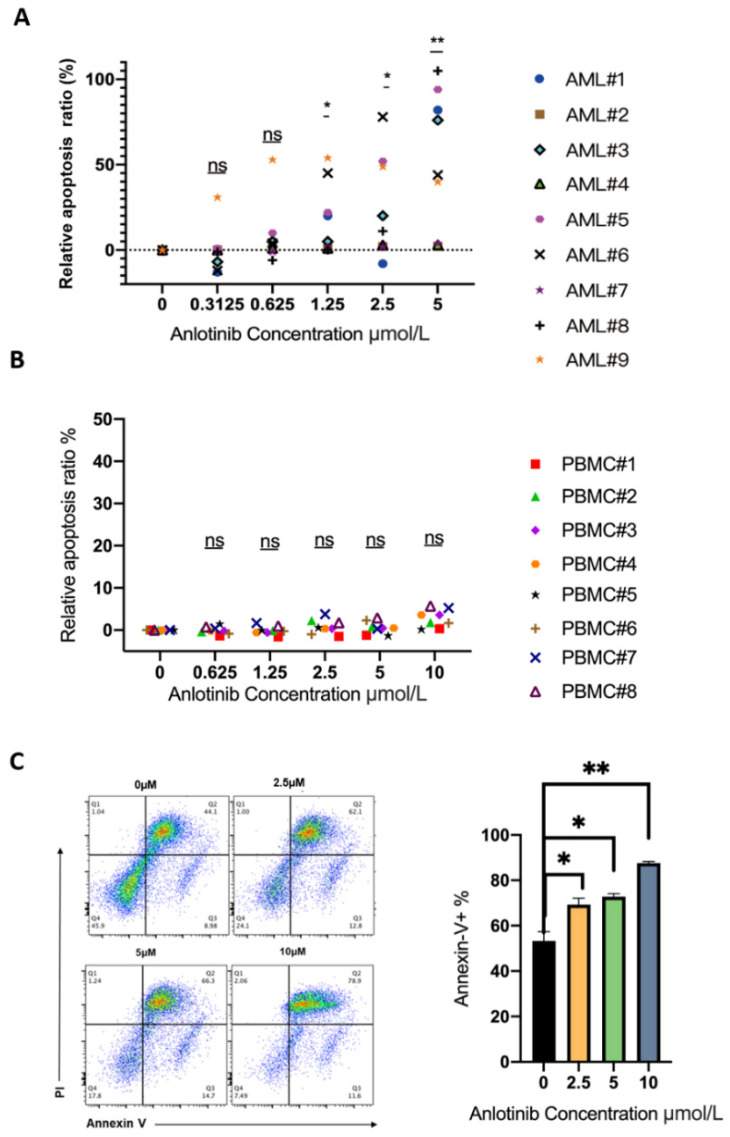
The cytotoxicity of anlotinib on primary CD34^+^ AML cells, normal leukocytes, and CD34^+^CD38^−^ LSCs ex vivo. (**A**) The relative apoptosis of CD34^+^ AML cells after anlotinib treatment with indicated concentration (*n* = 9). (**B**) The apoptosis of mononuclear cells from the peripheral blood of healthy donors (*n* = 8). (**C**) The apoptosis of CD34^+^CD38^−^ LSCs from AML patients (*n* = 3) * *p* < 0.05, ** *p* < 0.01.

**Figure 3 pharmaceuticals-15-01313-f003:**
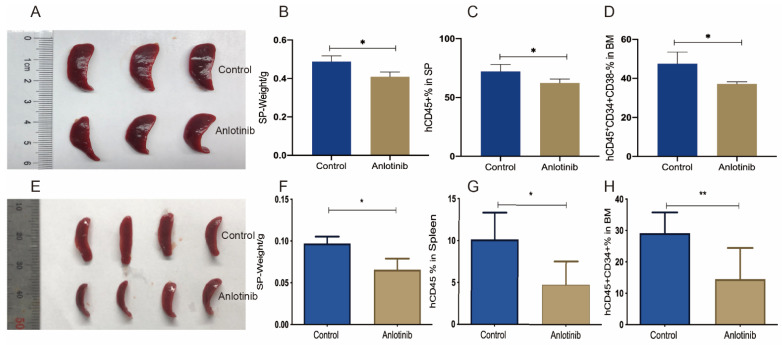
Efficacy of anlotinib in inhibiting regeneration of LSCs in PDX models. (**A**–**D**) Efficacy of anlotinib in impairing the regeneration capacity of CD34^+^CD38^−^LSCs in patient-derived leukemia xenograft mice model. The spleen size (**A**) and weight (**B**) and the percentages of hCD45^+^ cell in the spleen (**C**) and the percentages of CD45^+^CD34^+^ cells in bone marrow ((**D**), *n* = 3/group). (**E**–**H**) Efficacy of anlotinib in impairing the regeneration capacity of CD34^+^LSCs in patient-derived leukemia xenograft mice model. The spleen size (**E**) and weight (**F**) and the percentage of hCD45^+^ cell in the spleen (**G**) a and the percentage of CD45^+^CD34^+^ cells in bone marrow ((**H**), *n* = 4/group). * *p* < 0.05, ** *p* < 0.01.

**Figure 4 pharmaceuticals-15-01313-f004:**
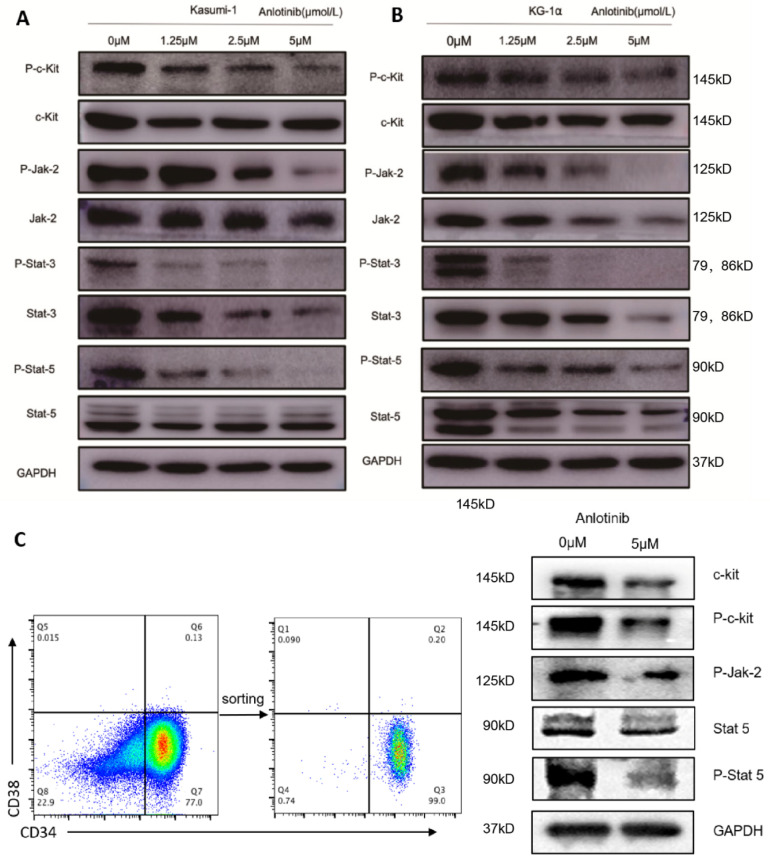
Effects of anlotinib on JAK/STAT signaling in LSC-like and CD34^+^CD38^−^LSC cells. (**A**,**B**) Western blot analysis of c-kit, p-c-kit, JAK2, p-JAK2, STAT3, p-STAT3, STAT5, and pSTAT5 in Kasumi-1 cells (**A**) and KG-1α cells (**B**) after anlotinib treatment with indicated concentration for 12 h. (**C**) Western blot analysis of c-kit, p-c-kit, p-JAK2, STAT3, p-STAT3, STAT5, and pSTAT5 after anlotinib treatment with indicated concentration for 12 h.

**Figure 5 pharmaceuticals-15-01313-f005:**
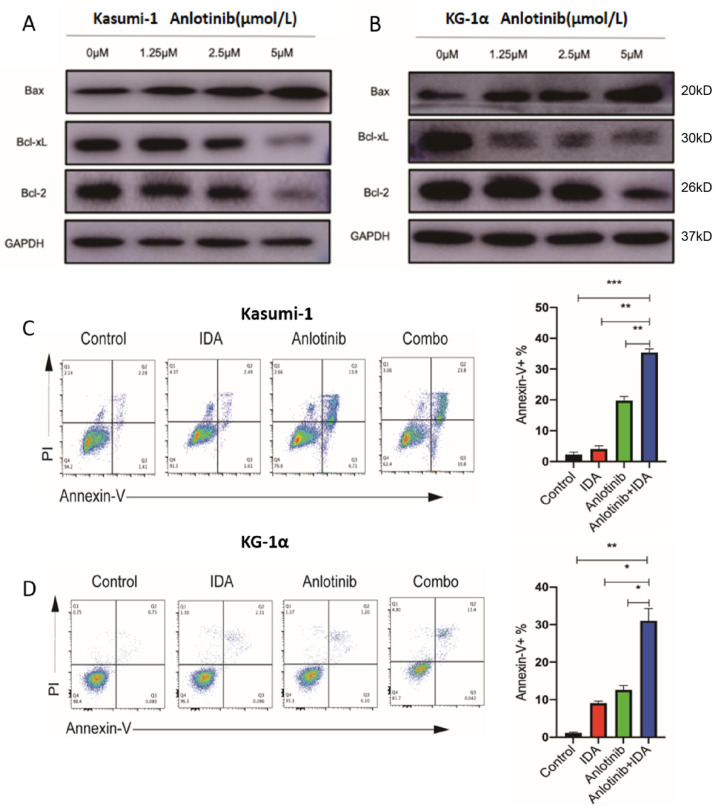
Effects of anlotinib on apoptotic-related proteins in LSC-like cells and its potential in enhancing chemosensitivity of LSC-like cells. (**A**,**B**) Western blot analysis of Bax, BCL-2, and BCL-xL in Kasumi-1 cells (**A**) and KG-1α cells (**B**) after anlotinib treatment with indicated concentration for 12 h. (**C**,**D**) Apoptosis of Kasumi-1 cells (**C**) and KG-1α cells (**D**) after monotherapy of anlotinib (2.5 μM), IDA (30 nM), and combinatory therapy of anlotinib (2.5 μM) and IDA (30 nM) for 48 h. * *p* < 0.05, ** *p* < 0.01, *** *p* < 0.001.

**Figure 6 pharmaceuticals-15-01313-f006:**
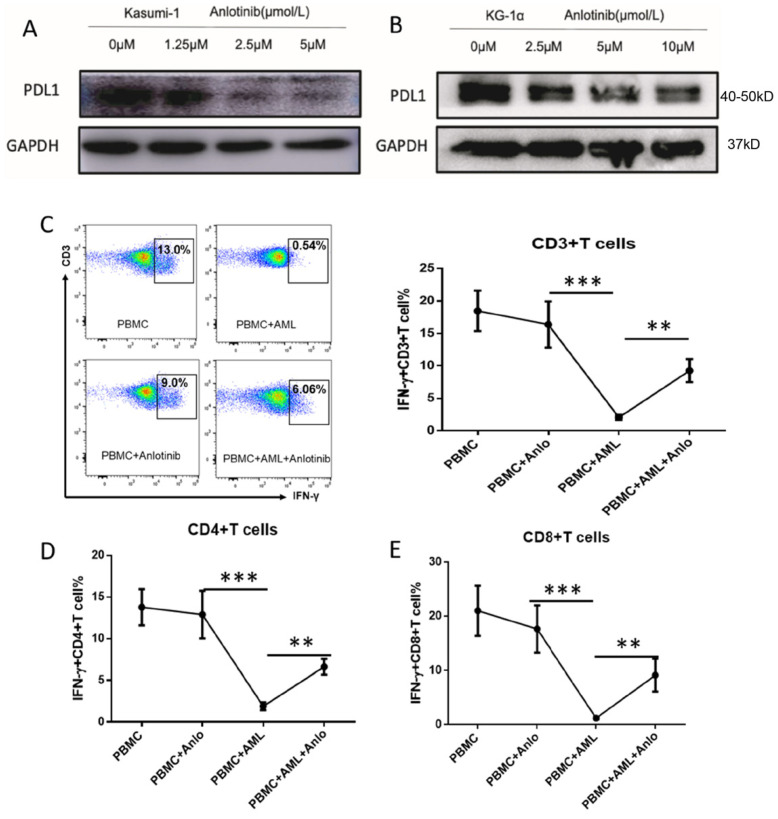
Effects of anlotinib on immune-evasion potential of LSC-like cells. (**A**,**B**) Western blot analysis of PD-L1 expression in Kasumi-1 cells (**A**) and KG-1α cells (**B**) after anlotinib treatment with indicated concentration for 12 h. (**C**,**D**) IFN-γ production in CD3^+^T cells (**C**), CD4^+^T cells, (**D**) and CD8^+^T cells (**E**) in the indicated groups (*n* = 6/group). ** *p* < 0.01, *** *p* < 0.001.

**Table 1 pharmaceuticals-15-01313-t001:** Clinical characteristics of AML patients.

Patient	Gender	Age	FAB	WBC (10^9^/L)	Blasts (%)	Cytogenetics	Molecular Mutations
1	M	39	M1	102.03	90.7	46,XY	ASXL1,IDH1,TET2
2	F	34	M5b	74	90.1	46,XX	NPM1,FLT3-ITD
3	M	51	M2	122	81.5	46,XY	CEBPA,TET2,ASXL1
4	M	45	M5	190.3	79.5	46,XY	FLT3-ITD, NPM1
5	M	38	M2a	29.3	46	46,XY	NRAS,PHF6,TET2,TP53
6	M	40	M2	10.75	69.3	46,XY	CEBPA,TAD1,bZIP
7	F	39	M5b	48.9	59.5	46,XX,t (9;11) (p22;q23)	MLL-AF9
8	M	56	M2b	44.4	21.7	45,XY,t(8;21)(q22;q22)	CEBPA,AML1-ETO
9	M	16	M2	41.7	81	46,XY	TET2, CEBPA

## Data Availability

Data is contained within the article.

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
