# Peer review of "Preclinical Evaluation of the Multiple Tyrosine Kinases Inhibitor Anlotinib in Leukemia Stem Cells"

_pharmaceuticals, 2022, doi:10.3390/ph15111313_

Round 1

Reviewer 1 Report

Yuelong Jiang and co-authors present a quality and well-written experimental manuscript describing preclinical evaluation of the multiple tyrosine kinases inhibitor anlotinib in leukemia stem cells.

Authors explore the role of anlotinib, a multiple tyrosine kinase inhibitor, in killing LSCs and regulating chemoresistance and immune evasion was explored. Anlotinib treatment induced apoptosis of LSC-like cells as well as primary AML LSCs, while sparing the normal mononuclear cells in vitro. The anti-leukemia activity of anlotinib was also confirmed in the mice model with Kasumi-1 cells without significant hematological and hepatic toxicities. Moreover, anlotinib was shown to impair the regeneration capacity of LSCs in the patient-derived leukemia xenograft mouse model. Mechanistically, anlotinib inhibited phosphorylation of c-kit, JAK2/STAT3 and STAT5, and downregulated STAT3 and STAT5 expression. In addition, anlotinib downregulated the anti-apoptotic proteins Bcl-2 and Bcl-xL, and upregulated Bax, thereby enhancing the sensitivity of LSCs to idarubicin in vitro. It also partially rescued the interferon-production of T cells cocultured with LSCs by downregulating PD-L1 expression.

Finally, authors conclude that their preclinical findings provided evidence for the clinical implementation of anlotinib as a novel approach to eliminate LSCs. Given the role of anlotinib in regulating chemosensitivity and immune evasion and its low adverse effects, combining chemotherapy such as IDA and cytarabine, or immunotherapy such as immune checkpoint inhibitors might be a promising strategy to improve the outcome of AML.

Overall, the manuscript is highly valuable for the scientific community and should be accepted for publication after the corrections are made.

==============================

Other comments:

1) Please check for typos throughout the manuscript.

2) With regards to patient mutations listed in Table 1 authors are kindly encouraged to cite the following article that describes various aspects of therapeutic targeting of mutant oncosuppressors, e.g p53. DOI: 10.3389/fonc.2020.01460

Author Response

  • Please check for typos throughout the manuscript.

Response:Thank you for your suggestion, we checked the typos and revised our manuscripts and marked in red.

  • With regards to patient mutations listed in Table 1 authors are kindly encouraged to cite the following article that describes various aspects of therapeutic targeting of mutant oncosuppressors, e.g p53. DOI: 10.3389/fonc.2020.01460

Response:Thank you for your comment; As requested, we cite the mentioned article as shown in line 331-332.

Reviewer 2 Report

Good work, I allow myself to put my remarks for a better presentation of your article 

1- How do you explain that the AML9 LSCs (figure 2C) are resistant to a high concentration of anlotinib

2- How are you sure that the selected AML LSCs are LSCs and not normal stem cells (NSCs) or contain a high concentration of NSCs

3- you must add as additional data the complete image of the western blot with weight marker and mention the weight of each targeted protein

4- in part 3.4 it is necessary to add the effect of Anlotinib in the chemosensitivity of primary LSC

Author Response

1-How do you explain that the AML9 LSCs (figure 2C) are resistant to a high concentration of anlotinib

Response:We appreciate your insightful comments. AML9 patients harbored both mutations of CEBPA AND TET2. Although CEBPA mutation generally indicates favorable outcome, However, recent report in EHA showed that cocurrence of CEBPA and TET2 mutations significantly compromise the outcome of AML patients by downregulating p53 pathways (doi: 10.1097 /01. HS9.0000843368. 69924.20). The disruption of p53 signaling might be contribute to resistance to apoptosis and anlotinib resistance. However, the detailed mechanism needs further exploration.

2- How are you sure that the selected AML LSCs are LSCs and not normal stem cells (NSCs) or contain a high concentration of NSCs

Response:Thank you for the valid suggestion. CD34+NSCs generally account for about 1% of bone marrow mononuclear cells in healthy donors.  In AML settings, normal stem cells expansion in bone marrow were significantly inhibited by AML cells, diagnosis of AML, AML cells generally should be ≥20%. Therefore, AML cells are at least 20 times of NSCs. In addition, the phenotype of LSC were considered as” CD34+CD38-“, AML cells of all patients we selected for LSC isolation were CD34 positive; therefore, NSCs could not account for high percentages in the study.  

3- you must add as additional data the complete image of the western blot with weight marker and mention the weight of each targeted protein

Response:Many thanks for your suggestion, we revised our manuscripts accordingly.

4- in part 3.4 it is necessary to add the effect of Anlotinib in the chemosensitivity of primary LSC.

Response:We appreciated for your valuable comment; Actually, it would be more convincible to explore the effect of anlotinib in the chemosensitivity of primary LSC; However, the deadline is too short for us to collect enough primary LSC samples. Additionally, the LSC-like cells:CD34+CD38-populations of Kasum-1 AND KG-1a were isolated and were used in the experiments in vitro, which might represent partial properties of LSCs. To avoid the inadequate evidence in primary LSCs, we revised the manuscript and highlighted the role in LSC-like cells rather than LSCs, and marked them in red in line 244 and 254.

Reviewer 3 Report

This is a well-designed and straightforward study of anlotinib efficacy for AML cells. Authors provide well controlled experiments which show antineoplastic anlotinib action in vitro and in vivo on leukemia LSC, explore potential mechanisms and show potential for anlotinib in combinational therapy with idarubicin. Although? As authors mention the durability of anlotinib action in vivo might be hindered by compensation mechanisms the main strength of this study is the use multiple AML and healthy donor samples, and relevant mouse model. There are only minor remarks which should be addressed before publishing:

1) Figure 1E and 1F- Sum for some bars does not equal 100%. For better comparison bars should be rescaled to 100%. Also why cell cycle was analyzed 6h after treatment and apoptosis after 48 and 72h? This seems confusing, as cell cycle changes usually take longer, and the rationale for 6h time point should be discussed in the text.

2) Figure 2A- What is a rationale for using relative apoptosis ratio y-scale for A panel compared to annexin V+ % scale for panel B? For better comparison both graphs should use either relative apoptosis ratio or annexin V+ %.

3) Figure 3A and 3E lack description which spleens are from treated/control groups.

4) Figure 3B-C- There is no reason to provide these graphs as split bars. These graphs might lead to misinterpretation of results, and should be provided with continuous scale, same as panels D-H. Also y-axis label for G is clipped.

5) For all western blots please provide protein marker sizes for each analyzed protein.

6) The statement that anlotinib has “potential to enhance the chemosensitivity” is to general, and cannot be concluded from experiments with idarubicin only. I recommend to rephrase it (especially in abstract) to highlight that anlotinib has potential to enhance the sensitivity to idarubicin and not to generalize on other chemotherapy.

7) This seems to be a continuation of work that showed that anlotinib suppresses MLL-rearranged AML via affecting SETD1A/AKT-mediated DNA damage response. However, in this manuscript I did not find mentions of this potential alternative mechanism and authors focus mainly on JAK2 and BCL2 signaling. I think some discussion can be added on potential mechanisms and whether SETD1A/AKT-mediated DNA damage might be relevant for AML LSC.

8) Some words are highlighted in red, seems like an artifact from manuscript editing (like lines 67, 73, 91).

9) Line 206- “BM of AML patients”- BM abbreviation is not described, I suppose it means bone marrow from healthy donors, but description for what BM means should be provided.

Author Response

1) Figure 1E and 1F- Sum for some bars does not equal 100%. For better comparison bars should be rescaled to 100%. Also why cell cycle was analyzed 6h after treatment and apoptosis after 48 and 72h? This seems confusing, as cell cycle changes usually take longer, and the rationale for 6h time point should be discussed in the text.

Response:We appreciate the carefulness of the reviewer; we revised the figure mentioned above;As shown in the figure 1A-D, large portions of AML cells were dead after 48h and 72h, which might interfere the results of cell cycle, especially in high dose group such as 2.5 and 5.0μM; Additionally, Kasumi-1 and KG-1a proliferate very fast, As shown in Fig1E-F, even treated by anlotinib, at least about 20% AML cells still underwent the process of proliferation. Therefore, we analyzed cell cycle at 6th h after treatment could reflect the cell cycle without influence of apoptosis.

2) Figure 2A- What is a rationale for using relative apoptosis ratio y-scale for A panel compared to annexin V+ % scale for panel B? For better comparison both graphs should use either relative apoptosis ratio or annexin V+ %.

Response:The suggestion is valid and constructive. We used the relative apoptosis rate in Fig2A to highlight the effect of anlotinib. However, the exact percentages of apoptosis is used to highlight the low toxicity of anlotinib in PBMCs in Fig2B. Actually, using both relative apoptosis ratio in A and B panel could display better comparison. We revised the the figure accordingly.

3) Figure 3A and 3E lack which spleens are from treated/control groups.

Response:We are sorry for the mistake; we revised the figure as requested.

4) Figure 3B-C- There is no reason to provide these graphs as split bars. These graphs might lead to misinterpretation of results, and should be provided with continuous scale, same as panels D-H. Also y-axis label for G is clipped.

Response:Many thanks for your kind suggestion, we revised the figures mentioned above accordingly.

5) For all western blots please provide protein marker sizes for each analyzed protein.

Response:The suggestion is important and meaningful, we revised all figures related to western blots as requested.

6) The statement that anlotinib has “potential to enhance the chemosensitivity” is to general, and cannot be concluded from experiments with idarubicin only. I recommend to rephrase it (especially in abstract) to highlight that anlotinib has potential to enhance the sensitivity to idarubicin and not to generalize on other chemotherapy.

Response:We really appreciate the precise and valuable suggestion. we revised the manuscript accordingly and marked in red in line 47,87,244,253,272 and 322.

7) This seems to be a continuation of work that showed that anlotinib suppresses MLL-rearranged AML via affecting SETD1A/AKT-mediated DNA damage response. However, in this manuscript I did not find mentions of this potential alternative mechanism and authors focus mainly on JAK2 and BCL2 signaling. I think some discussion can be added on potential mechanisms and whether SETD1A/AKT-mediated DNA damage might be relevant for AML LSC.

Response:The comments are insightful. Indeed, our group explored the role of anlotinib in MLL-rearranged AMLvia SETD1A/AKT signaling pathway. The study was performed based on the special feature of MLL-rearranged AML and the important role of SETD1A in those cell lines (Cell. 2018 Feb 22; 172(5): 1007–1021.e17). However, we tried to explore the role of anlotinib in LSCs regardless of the genetic background due to the inhibitory effects in multiple tyrosine kinases. Since the dependence of growth signaling and the activation of the downstream JAK2 signaling in LSCs, we focused on the JAK2 signaling and its target BCL2 when exploring mechanisms. As our group and the current study did not refer to data about SETD1A/AKT-mediated DNA damage in LSCs, discussion about the issue can be hardly performed and might be considered as over-speculative. However, your constructive suggestion inspires us to explore whether anlotinib affecting SETD1A/AKT-mediated DNA damage in LSCs in future.

8) Some words are highlighted in red, seems like an artifact from manuscript editing (like lines 67, 73, 91).

Response:We really appreciate the reviewer’s kind remainder and carefulness; we revised the manuscript as requested.

9) Line 206- “BM of AML patients”- BM abbreviation is not described, I suppose it means bone marrow from healthy donors, but description for what BM means should be provided

Response:We apologize for the negligence; we added the description of BM in Line 207 accordingly.

Round 2

Reviewer 2 Report

Dear,

you must add as additional data the complete image of the western blot with weight marker and mention the weight of each targeted protein
